# Belowground Response of a Bahiagrass Pasture to Long-Term Elevated [CO_2_] and Soil Fertility Management

**DOI:** 10.3390/plants13040485

**Published:** 2024-02-08

**Authors:** G. Brett Runion, Stephen A. Prior, H. Allen Torbert

**Affiliations:** United States Department of Agriculture-Agricultural Research Service, National Soil Dynamics Laboratory, 411 S. Donahue Drive, Auburn, AL 36832, USA; steve.prior@usda.gov (S.A.P.); allen.torbert@usda.gov (H.A.T.)

**Keywords:** carbon dioxide, pasture, fertilization, global change, roots, rhizomes

## Abstract

Effects of rising atmospheric CO_2_ concentration [CO_2_] on pastures and grazing lands are beginning to be researched, but these important systems remain understudied compared to other agronomic and forest ecosystems. Therefore, we conducted a long-term (2005–2015) study of bahiagrass (*Paspalum notatum* Flüggé) response to elevated [CO_2_] and fertility management. The study was conducted at the USDA-ARS, National Soil Dynamics Laboratory open-top field chamber facility, Auburn, AL. A newly established bahiagrass pasture was exposed to either ambient or elevated (ambient + 200 µmol mol^−1^) [CO_2_]. Following one year of pasture establishment, half the plots received a fertilizer treatment [N at 90 kg ha^−1^ three times yearly plus P, K, and lime as recommended by soil testing]; the remaining plots received no fertilization. These treatments were implemented to represent managed (M) and unmanaged (U) pastures; both are common in the southeastern US. Root cores (0–60 cm depth) were collected annually in October and processed using standard procedures. Fertility additions consistently increased both root length density (53.8%) and root dry weight density (68.2%) compared to unmanaged plots, but these root variables were generally unaffected by either [CO_2_] or its interaction with management. The results suggest that southern bahiagrass pastures could benefit greatly from fertilizer additions. However, bahiagrass pasture root growth is unlikely to be greatly affected by rising atmospheric [CO_2_], at least by those levels expected during this century.

## 1. Introduction

The unprecedented, well-documented increase in atmospheric carbon dioxide concentration [CO_2_] can be primarily attributed to anthropogenic activities such as fossil fuel burning and land use change [1]. It is well known that elevated [CO_2_] can increase photosynthesis and resource use efficiencies, leading to increased plant growth; however, the magnitude of the response depends on differences in photosynthetic pathways among plant species [2,3]. Plants with a C_3_ photosynthetic pathway (e.g., soybean, cotton) exhibit both increased water use efficiency and photosynthesis; C_4_ plants (e.g., corn, sorghum) also show increased water use efficiency but a lower photosynthetic response due to their CO_2_-concentrating mechanism [2,3,4,5,6,7]. As a result, C_3_ plants often show a much greater biomass response to elevated [CO_2_] (33–40% increase) than C_4_ plants (10–15% increase) [3,8,9,10].

When soil resources such as N are limiting, plants may not respond to increased atmospheric [CO_2_] [11,12]. Nitrogen is the element most limiting to biomass production [13,14] and is key to both plant and soil C dynamics.

Rangelands and pastures represent systems where soil N is often limiting, indicating that an increase in plant production from rising [CO_2_] may require soil N addition [15]. These systems also represent a type of conservation system in that they are not tilled and have an ability to sequester carbon in roots and soil. Knowledge on how these systems respond to future environmental conditions will be important since they will impact both grazing and haying operations important to animal production. While the effects of elevated [CO_2_] on rangelands have received some attention, pastures—particularly those in the southeastern US—remain an understudied agroecosystem [15]. Bahiagrass (*Paspalum notatum* Flüggé), native to South America and introduced into the U.S. in 1913 [16], is a common and important C_4_ grass in southeastern US pastures [17,18,19]. It has been suggested that production increases could occur on grazing lands with increased atmospheric [CO_2_] in humid temperate regions such as the southeastern US [20], where pastures occupy 32.4 million ha, about 75% of the total pasture area in the eastern US [21]. Accurately determining the total acreage of bahiagrass is difficult since pasture statistics show acreage combined across forage types. However, it is known that bahiagrass is the predominant forage for the beef cattle industry in the southeastern US [17] and was estimated to cover at least 2.5 million hectares [22]. In the same 10-year bahiagrass study described here, Prior et al. [23] found a consistent strong (>100% increase) aboveground biomass response to fertilizer additions. The aboveground response to elevated [CO_2_] was less consistent and much lower (13.8% increase), albeit in line with other studies with C_4_ species [8]. They further observed that a positive growth response to elevated [CO_2_] was only observed in the fertilized plots, indicating that future pasture management should consider fertilization to take advantage of rising atmospheric [CO_2_] levels.

Although effects of elevated [CO_2_] on roots have been much less studied than aboveground plant organs, they frequently show positive growth responses such as length, weight, and branching [24,25,26,27]. In fact, in some studies roots have shown the greatest relative (compared to ambient growth conditions) dry weight gain among organs when exposed to elevated [CO_2_] [10,28,29]. The possible implications of increased rooting from elevated [CO_2_] and/or fertility on plants are varied and include the following: increased exploration for water and nutrients [30]; a greater ability of plants to withstand periods of biotic or abiotic stress [31,32,33]; and increased symbiosis with beneficial microorganisms [34]. In addition to effects on plants, increased rooting from elevated [CO_2_] may also impact soil processes such as C sequestration [35,36,37], soil CO_2_ efflux [38,39], N availability [40], water infiltration and water holding capacity [41], and N leaching to groundwater, with its associated health risks [30].

Understanding the interactions of fertility management with rising [CO_2_] on plant/soil systems will be crucial to the management of these systems for both profitable and environmentally sound agricultural systems of the future [42,43]. Clearly, more research is needed to fully understand these complex interactions.

To begin to fill this knowledge gap, a long-term experiment examining above- and belowground responses of an important southeastern pasture system (bahiagrass) to ambient and elevated levels of [CO_2_] with a fertility management treatment (no fertilizer = unmanaged and added fertilizer = managed) was initiated. The aboveground responses have been previously published [23]; this manuscript focuses on the long-term belowground responses of this bahiagrass pasture to elevated [CO_2_] and fertility management. We hypothesized that a [CO_2_] response would be observed only in managed plots. This 10-year study was implemented using open-top chambers on a soil bin located at the USDA-ARS National Soil Dynamics Laboratory in Auburn, AL.

## 2. Results

### 2.1. Root Length Density

Overall (across all 10 yrs and all six depths), RLD was significantly (*p* < 0.001) increased under M (54%) but was unaffected by [CO_2_] (Table 1). The interaction of these treatments was also not significant since RLD was greater under M in both A and E [CO_2_] treatments, while [CO_2_] had no effect on RLD under either M or U.

Further, the effects of depth across all 10 yrs showed that M increased RLD at all depths except at 0–5 cm (Table 2). Similar to the overall results, RLD was unaffected by [CO_2_] at all depths and interactions were also not significant due to M increasing RLD at all depths (except 0–5 cm) in both A and E, while [CO_2_] did not affect RLD in either M or U systems at any depth.

When examined across all six depths, RLD showed similar patterns in each of the 10 years; that is, RLD under M was greater than U in most years with 2012, 2014, and 2015 exhibiting trends (Table 3). Further, there was no main effect of [CO_2_] in any year. Interactions followed the general pattern of not being significant due to RLD being greater under M than U in both A and E in all years and A not being different than E in both M and U in all years.

Examination of RLD for all years and at all depths showed that M increased RLD compared to U at most depths in most years (Table 4). Exceptions occurred at the 0–5 cm depth in 2008 and 2011–2015 where M and U were not significantly different. Further, in 2015 M and U did not differ significantly at the 10–15, 15–30, and 30–45 cm depths. The main effect of [CO_2_] on RLD was generally not significant for most depths in most years. Although infrequent, when a significant effect was noted, RLD in E was generally higher than A except at the 45–60 cm depth in 2007 and 2009. The interaction of M x [CO_2_] was also generally not significant. As noted in previous interactions, M was greater than U in both A and E, while A and E did not differ in both M and U. When a significant interaction was noted, A and E differed only under M; however, there was no consistent pattern with regard to whether A or E had greater RLD under M. For example, A was greater than E under M at the 5–10, 15–30, and 30–45 cm depths in 2007, while E was greater than A in M at 30–45 and 45–60 cm depths (similar trends at 10–15 and 15–30 cm) in 2011.

### 2.2. Root Weight Density

When examined across all years and depths, overall RWD followed a pattern similar to RLD in that there was a strong effect of soil fertility management (i.e., M greater than U) while the effect of [CO_2_] was not significant (Table 1). Across years and depths, RWD was significantly (*p* < 0.001) increased under M (68%). The interaction of these treatments was not significant; as with RLD, RWD was increased by M in both A and E [CO_2_] treatments, while [CO_2_] had no effect on RWD under either M or U.

Across all 10 years, M increased RWD at all depths (Table 2). Further, RWD was unaffected by [CO_2_] at all depths. Interactions showed that M increased RWD at all depths in both A and E, while [CO_2_] did not affect RWD under either M or U at any depth. As before, this resulted in these interactions not being statistically significant.

When examined across all six depths, RWD showed similar patterns to RLD in each of the 10 years; that is, RWD under M was greater than U in all years (trend in 2012; Table 3). Again, there was no significant main effect of [CO_2_] in any year. Interactions followed the general pattern of not being significant due to RWD being greater under M than U in both A and E in all years and A not being different than E in both M and U in all years.

When examined for all years and at all depths, RWD was increased under M (vs. U) at most depths in most years (Table 5). Exceptions occurred at the 0–5 cm depth (2008, 2012, 2013, and 2015) and the 5–10 cm depth (2014), where M and U were not significantly different. However, for 2014 (5–10 cm) and 2015 (0–5 cm), M showed a trend of being higher. The main effect of [CO_2_] on RWD was generally not significant for most depths in most years. Although infrequent, significant effects of [CO_2_] generally showed RWD was higher in E than A. The interaction of soil fertility management with [CO_2_] was also generally not significant. Significant interactions were only noted in 2009 and 2011; these showed that A and E differed only in the managed system, E usually had greater RWD than A, and differences only occurred at or below the 10–15 cm depth.

### 2.3. Lineal Root Density

Overall, across all years and all depths, LRD was not significantly affected by soil fertility management or [CO_2_] (Table 1). Lineal root density was also not influenced by the interaction of these factors.

When examined by depth across years, M significantly increased LRD only at the 0–5 cm depth (Table 2). Lineal root density was unaffected by [CO_2_] or by the interaction of soil fertility management with [CO_2_].

When examined by year across all six depths, LRD was greater under M compared to U in 2008–2010, with a similar trend noted in 2007 (Table 3). During these years, LRD varied from 13.57–16.54 g km^−1^ in M and from 11.73–14.20 g km^−1^ in U. In 2013, an opposite pattern in LRD was noted in that U was higher than M. For all other years, the effect of soil fertility management on LRD was not significant. Effects of [CO_2_] and the interaction of soil fertility management with [CO_2_] on LRD were not significant in any year when examined across all six depths.

For all years and depths, soil fertility management had infrequent effects on LRD and was only significant in 14 of the 60 possible years by depth combinations (Table 6). In general, higher LRD values under M tended to occur early in the study (i.e., 2007–2011). In comparison, the U treatment had higher LRD from 2012–2015. In both cases, the depths at which these differences were noted varied with no discernible patterns. The main effect of [CO_2_] on LRD was significant only at 0–5 cm (E > A) and 15–30 cm (A > E) in 2011, 5–10 cm (E > A) in 2013, and 10–15 cm (E > A) in 2014. The interaction of soil fertility management with [CO_2_] was almost always not significant, with no discernible pattern on the rare occurrences when it was significant.

### 2.4. Rhizome Dry Weight and Root-to-Shoot Ratio

Soil fertility management increased rhizome dry weight (~68%), following the same general pattern noted for most root data (Table 7). There was also somewhat of a trend (*p* = 0.136) for elevated [CO_2_] to increase (13.2%) rhizome dry weight. There was no significant interaction for this variable.

Compared to managed conditions, the root-to-shoot ratio (R:S) was much higher (1.73) for bahiagrass grown under unmanaged conditions where no fertilizer additions had been made (Table 7). The main effect of [CO_2_] and the management by [CO_2_] interaction were both not significant for R:S.

## 3. Discussion

The relatively positive effects of fertilizer addition on both RLD and RWD in the managed plots were expected and are common in these types of studies. It is logical, given that plants are unable to build tissue, including roots, without sufficient resources such as required nutrients. The fact that the response to management was much larger (>100% increase for M over U) for aboveground tissue [23] than for roots (mean response ~54%) was somewhat unexpected. This differential response could possibly be explained by Liebig’s law of the minimum (see, for example, [44]) in that, once the bahiagrass had produced enough roots to provide sufficient belowground resources (water and nutrients) to build tissue, the plants allocated more resources to collecting aboveground resources such as light and CO_2_ for photosynthesis [45]. While it has been suggested that Liebig’s law is simplistic in its relationship to crop yield since there are complex interacting factors affecting yield [46], it is still adequate to explain the differential response observed between the above- and belowground responses observed in this study. The large response in root production to pasture management indicates that growers should consider nutrient additions to increase yields.

While the effects of fertilization were more or less expected, the fact that [CO_2_] had little overall impact on bahiagrass root growth was unexpected. Past research has generally shown plants have a significant positive belowground response to growth in elevated [CO_2_]. Rogers et al. [26] showed that elevated [CO_2_] increased root growth (especially RDWD) in ~87% of studies, regardless of study conditions. While C_4_ plants generally show a lower response to elevated [CO_2_] than C_3_ plants [2], even C_4_ plants usually show a positive growth response to elevated [CO_2_] [26]. Although there are studies that have not observed a positive root growth response to elevated [CO_2_] [26], these have usually been conducted in growth chambers, phytotrons, or greenhouses with plants growing in containers, which might limit the belowground response to elevated [CO_2_] [47]. However, even work in OTCs has shown a lack of response to [CO_2_]. Curtis et al. [48] found that roots of the C_3_ sedge *Scirpus olneyi* increased in OTCs under elevated [CO_2_], but the C_4_ grass *Spartina patens* did not; also in a mixed community, roots of the C_4_ grass *Distichlis spicata* also did not respond to elevated [CO_2_]. They suggested that young plants often show stronger responses to elevated [CO_2_], while the mature marsh communities in their study responded less (or not at all in the case of the C_4_ plants). It is unknown why the bahiagrass roots did not show a positive response to elevated [CO_2_] in the current study, but it supports the contention of Curtis et al. [48] in that this was—for most of a decade—a mature, C_4_ pasture community.

In addition to there being virtually no effect of [CO_2_] on root growth variables, there were no generalized effects of the interaction of [CO_2_] and management on these measured variables. As with the lack of a [CO_2_] response, this result was unexpected. It is not uncommon for plants to respond to elevated [CO_2_] only when other resources (fertility and water) are adequate enough to provide the plant the ability to utilize the added [CO_2_] to build additional biomass. This was generally observed for the aboveground tissue in this study ([23]; elevated > ambient in M, but not U) and has been observed in other studies at our facility ([12]; elevated > ambient in high N but not low N). As with [CO_2_], it is unknown why this typical [CO_2_] fertility pattern was not observed for roots in the current study, but several factors (plant species, soil type, CO_2_ level tested, timing and method of root collection, etc.) may have impacted this result.

Adding nutrients increased rhizome dry weight in similar proportions to RWD. Plants grown in elevated [CO_2_] showed a slight increase in rhizome dry weight, which is in alignment with general responses of C_4_ plants to elevated [CO_2_]. The fact that rhizomes are part aboveground structure—which tended to show positive responses to elevated [CO_2_]—and part belowground structure—which generally did not respond to elevated [CO_2_]—makes this trend a somewhat interesting “average” response. The R:S of the unmanaged plants was more than twice that of those under fertility management. This indicates that unmanaged bahiagrass pastures are expending resources to build roots much more than aboveground tissues and that these plants are exploring the soil for needed, albeit absent, resources. These plants should be “primed” (with roots in place) to take advantage of any nutrient addition should the management practices change. There was no significant effect of [CO_2_] on R:S. A review by Rogers et al. [29] showed that elevated [CO_2_] had no effect on R:S in only 3% of studies (compared to 58% showing increased R:S and 38% showing decreased R:S) but added that this response is highly variable among crop species and experimental conditions. Given that aboveground dry weights in this study were generally increased by elevated [CO_2_] [23] while roots were unaffected, one might expect R:S to decline. The fact that this was not seen is likely due to the fact that aboveground data from the October harvest—which showed lower harvested dry weights than June and August harvests—were used in the calculation, resulting in a large variability in the data. The fact that R:S was also unaffected by the interaction of management with [CO_2_] resulted from the very large increase in R:S in unmanaged plots in both ambient and elevated [CO_2_].

## 4. Material and Methods

A bahiagrass pasture was established at the USDA-ARS National Soil Dynamics Laboratory (Auburn, AL, USA) in spring 2004 on an outdoor soil bin (7 m × 76 m × 2 m deep) containing Blanton loamy sand (loamy, siliceous, thermic Grossarenic Paleudult) supported on a tile and gravel drainage basin [49]. The soil was rototilled to a depth of 50 cm and bahiagrass seed was sown at a rate of 28 kg ha^−1^; additional pasture establishment procedures concerning soil testing and fertilization have been previously described [23]. In addition, weather data for the 10-year study period, which included maximum, minimum, and average daily temperatures (°C), precipitation amounts (mm), and 30-year averages for these weather variables, were previously reported by Prior et al. [23].

On 8 December 2004, atmospheric [CO_2_] exposure treatments were initiated and included ambient and elevated (ambient + 200 µmol mol^−1^) atmospheric [CO_2_]. Treatments were applied using structural aluminum-framed (3.05 m diameter by 2.40 m height) open-top field chambers (OTCs) covered with 0.2 mm PVC film panels [50]. Twelve OTCs were used in this study (six at ambient [CO_2_] and six at elevated [CO_2_]). For the entire 10-year study period, elevated [CO_2_] exposures were conducted during daylight hours (12 hr d^−1^). The utilized [CO_2_] monitoring and dispensing system was previously described by Mitchell et al. [51]. Briefly, a 12.7 Mg liquid receiver supplied CO_2_ that was dispensed through a high-volume manifold and continuously injected into fan plenum boxes that were connected to the bottom half of the double-walled PVC film panel. The inside wall of these panels was perforated with 2.5-cm diameter holes, which served as ducts for uniform air distribution into the OTCs. Fans were used, which ensured that three chamber volumes of air were exchanged every minute. A time-shared manifold with samples drawn through solenoids to an infrared CO_2_ analyzer (Li-Cor 6252, Li-Cor, Inc., Lincoln, NE, USA) was used to monitor [CO_2_] 24 hr d^−1^; each chamber was assessed for one minute per cycle, which included calibration gases (at a known [CO_2_] and N as a zero). A datalogger (CR-10, Campbell Scientific, Inc., Logan, UT, USA) was used to record [CO_2_] data, which were readily available for real-time observation on a dedicated computer. Average [CO_2_] (±SE) were 407.64 ± 0.04 (n = 381,639) and 599.02 ± 0.11 µmol mol^−1^ (n = 376,326) for the ambient and elevated [CO_2_] treatments, respectively. Over the 10-year study, 91.2% of recorded elevated [CO_2_] were within ±20% of our target value (ambient + 200 µmol mol^−1^).

On 25 April 2006, soil fertility management treatments (no fertilizer added vs. fertilizer added) were initiated to represent unmanaged (U) and managed (M) pastures, respectively; both systems are common in the southeastern US. Nitrogen was applied according to extension recommendations to M plots only. It is important to note that, since N is most limiting to forage production, this primary nutrient was applied three times per year (April, June, and August) at 90 kg ha^−1^ per application (total = 270 kg N ha^−1^ yr^−1^) as ammonium sulfate ([(NH_4_)_2_SO_4_]; N-P-K = 21-0-0). Other aspects of soil fertility (other than N) were also followed based on extension soil test recommendations in M areas only. In this regard, P as triple super phosphate (P_2_O_5_; N-P-K = 0-46-0) and K as muriate of potash (KCl; N-P-K = 0-0-60) were applied in April of each year at typical rates of 45 and 67.5 kg ha^−1^, respectively. During this same period, soil test recommendations required lime to be applied at 3363 kg ha^−1^ in 2007 and 2008 and at 2242 kg ha^−1^ in 2015 in M areas only. While we recognize that lime is not considered a fertilizer (although it is a source of calcium and magnesium) and is generally considered a soil amendment or pH adjustment treatment, since lime was only added three times during this study, we will continue to refer to this as a fertility management treatment. Unmanaged pasture areas received no fertilizer or lime during the 10-year study period.

A split-plot design was used in this study. The soil bin was divided lengthwise into three blocks (7 m wide × 25.33 m long); each block contained 4 OTCs for a total of 12 OTCs. Soil fertility management treatments [managed (M) or unmanaged (U)] were randomly assigned to one-half of each block and represented the whole plot treatment. Within each management treatment in each block, atmospheric [CO_2_] treatments [ambient (A) or elevated (E)] were randomly assigned to OTCs and represented the split-plot treatment.

Belowground biomass was assessed in October of each year of study (2006–2015) by extracting four soil cores (3.8 cm dia. × 60 cm long) for determination of fine root length density (RLD; km m^−3^), root dry weight density (RWD; kg m^−3^), and lineal root density (g km^−1^). Cores were extracted from each chamber using the methods described by Prior and Rogers [52] and stored at 4^o^ C until processing. Care was taken to ensure soil cores were not collected from previously sampled areas through use of a study-long sampling grid map. Cores were divided into 6 depth increments (0–5, 5–10, 10–15, 15–30, 30–45, and 45–60 cm); roots were extracted using a hydropneumatic elutriation system (Gillison’s Variety Fabrication, Inc., Benzonia, MI, USA; [53]) and stored in 70% ethanol [54] at 4 °C until processing. After organic debris was removed with tweezers and spring-loaded suction pipettes, root length was measured with a Comair Root Length Scanner (Hawker de Havilland, Port Melbourne, Australia). Root weight determinations were made after drying samples at 55 °C to a constant mass. In addition to root cores, belowground rhizomes were collected at study termination in October 2015. Sharpshooter spades were used to excavate rhizomes from soil for the entire chamber area. Rhizomes were separated from soil using the sieve method [54]. Rhizome weight determinations were made after drying samples at 55 °C to a constant mass. Belowground biomass (root + rhizome mass in October 2015), in combination with aboveground biomass from the October 2015 harvest reported by Prior et al. [23], was used to calculate the root-to-shoot ratio (R:S).

Data analysis was conducted using the Mixed Models Procedure (Proc Mixed) of SAS [55]; the data presented in all tables are means derived from this analysis. Error terms appropriate to the split-plot design were used to test the significance of main effects and their interactions. A significance level of (*p* ≤ 0.10) was established a priori.

## 5. Conclusions

The results from this study indicate that bahiagrass pastures in the southeastern US could benefit greatly from nutrient management. While fertilizer additions increased both root length density and root dry weight density compared to unmanaged areas, all assessed root variables were unaffected by either [CO_2_] or its interaction with fertility management. These results suggest that bahiagrass pasture root growth is unlikely to be greatly affected by rising atmospheric [CO_2_], at least by those levels expected during this century. However, the higher R:S in unmanaged areas suggests that unmanaged pastures could be primed to take advantage should nutrients be added in the future. Additional research with other pasture species and in other areas is required to determine if the findings presented here can be generalized across plant species or across areas of the southeastern US; some of this research is currently ongoing at our research facility.

## Figures and Tables

**Table 1 plants-13-00485-t001:** Bahiagrass root variables across all 10 years (2006–2015) and 6 depths (0–60 cm). Data shown are means (N = 180; 10 years*6 depths*3 reps) with statistics.

Trt ^a^	Root Length Density (km m^−3^)	Root Weight Density (kg m^−3^)	Lineal Root Density (g km^−1^)
**AU**	93.19	1.29	15.88
**AM**	145.12	2.23	16.46
**EU**	96.51	1.34	15.81
**EM**	146.71	2.20	16.17
**M ^b^**	<0.001	<0.001	0.247
**CO_2_**	0.740	0.897	0.522
**MxCO_2_**	0.907	0.694	0.694

^a^ Treatments = ambient CO_2_-unmanaged (AU); ambient CO_2_-managed (AM); elevated CO_2_-unmanaged (EU); elevated CO_2_-managed (EM). ^b^ *p* values for the main effects of management (M), carbon dioxide (CO_2_), and their interaction (MxCO_2_).

**Table 2 plants-13-00485-t002:** Bahiagrass root variables by depth across all 10 years (2006–2015). Data shown are means (N = 30; 10 years*3 reps) with statistics.

	**Root Length Density** **(km m**^**−3**^)					
**Trt** ^**a**^	**0–5 cm**	**5–10 cm**	**10–15 cm**	**15–30 cm**	**30–45 cm**	**45–60 cm**
**AU**	314.58	90.04	54.92	43.27	41.38	38.92
**AM**	365.92	192.23	108.43	85.62	66.82	82.55
**EU**	337.34	88.07	57.65	42.48	40.74	38.59
**EM**	377.21	187.24	113.14	88.99	67.72	77.64
**M ^b^**	0.183	<0.001	0.001	0.001	0.005	0.002
**CO_2_**	0.354	0.610	0.410	0.720	0.965	0.520
**MxCO_2_**	0.754	0.825	0.826	0.562	0.792	0.573
	**Root Weight Density (kg m^−3^)**					
**Trt ^a^**	**0–5 cm**	**5–10 cm**	**10–15 cm**	**15–30 cm**	**30–45 cm**	**45–60 cm**
**AU**	3.70	1.34	0.85	0.70	0.71	0.72
**AM**	4.98	2.88	1.68	1.41	1.25	1.54
**EU**	4.10	1.29	0.92	0.68	0.70	0.66
**EM**	5.01	2.88	1.75	1.43	1.16	1.39
**M**	<0.001	<0.001	<0.001	<0.001	0.008	0.015
**CO_2_**	0.461	0.812	0.282	0.958	0.398	0.176
**MxCO_2_**	0.513	0.852	0.955	0.723	0.486	0.558
	**Lineal Root Density (g km^−1^)**					
**Trt ^a^**	**0–5 cm**	**5–10 cm**	**10–15 cm**	**15–30 cm**	**30–45 cm**	**45–60 cm**
**AU**	11.76	15.01	16.26	16.59	17.65	19.03
**AM**	13.93	15.22	15.97	16.58	17.70	18.62
**EU**	11.91	15.15	16.82	16.67	17.80	17.59
**EM**	13.68	15.53	16.00	16.48	17.42	18.30
**M**	0.036	0.710	0.502	0.883	0.676	0.833
**CO_2_**	0.922	0.715	0.718	0.995	0.479	0.212
**MxCO_2_**	0.724	0.883	0.745	0.891	0.370	0.425

^a^ Treatments = ambient CO_2_-unmanaged (AU); ambient CO_2_-managed (AM); elevated CO_2_-unmanaged (EU); elevated CO_2_-managed (EM). ^b^ *p* values for the main effects of management (M), carbon dioxide (CO_2_), and their interaction (MxCO_2_).

**Table 3 plants-13-00485-t003:** Bahiagrass root variables by year across all 6 depths (0–60 cm). Data shown are means (N = 18; 6 depths*3 reps) with statistics.

	**Root Length Density (km m**^**−3**^)				
**Trt** ^**a**^	**2006**	**2007**	**2008**	**2009**	**2010**
**AU**	52.70	56.42	96.68	97.01	101.96
**AM**	88.70	120.60	138.82	159.75	166.24
**EU**	52.46	65.68	109.25	100.09	99.99
**EM**	94.19	109.08	137.27	176.47	162.81
**M ^b^**	0.001	0.003	0.056	0.011	0.006
**CO_2_**	0.819	0.949	0.762	0.712	0.905
**MxCO_2_**	0.803	0.556	0.697	0.799	0.974
	**2011**	**2012**	**2013**	**2014**	**2015**
**AU**	112.27	115.84	98.30	104.99	95.69
**AM**	170.40	170.52	154.30	140.56	141.34
**EU**	113.36	127.59	91.26	97.26	108.17
**EM**	188.93	159.33	155.97	139.13	143.98
**M**	0.017	0.144	0.013	0.142	0.107
**CO_2_**	0.721	0.992	0.910	0.861	0.763
**MxCO_2_**	0.750	0.697	0.855	0.904	0.844
	**Root Weight Density (kg m^−3^)**				
**Trt**	**2006**	**2007**	**2008**	**2009**	**2010**
**AU**	0.72	0.74	1.31	0.92	1.07
**AM**	1.35	1.82	1.98	1.99	2.34
**EU**	0.74	0.78	1.33	1.01	1.07
**EM**	1.40	1.61	2.04	2.09	2.21
**M**	<0.001	<0.001	<0.001	<0.001	<0.001
**CO_2_**	0.805	0.650	0.852	0.670	0.780
**MxCO_2_**	0.931	0.503	0.924	0.987	0.767
	**2011**	**2012**	**2013**	**2014**	**2015**
**AU**	1.42	1.96	1.75	1.61	1.44
**AM**	2.39	2.82	2.75	2.45	2.37
**EU**	1.45	2.18	1.72	1.57	1.58
**EM**	2.71	2.67	2.69	2.41	2.20
**M**	0.001	0.130	0.007	0.021	0.011
**CO_2_**	0.602	0.932	0.895	0.919	0.936
**MxCO_2_**	0.689	0.679	0.968	0.992	0.602
	**Lineal Root Density (g km^−1^)**				
**Trt**	**2006**	**2007**	**2008**	**2009**	**2010**
**AU**	14.57	14.59	14.74	11.31	11.86
**AM**	15.51	16.80	15.53	13.97	15.12
**EU**	15.14	13.81	13.56	12.15	11.75
**EM**	15.28	16.28	16.28	13.16	14.86
**M**	0.431	0.102	0.008	0.075	0.064
**CO_2_**	0.770	0.337	0.738	0.982	0.729
**MxCO_2_**	0.482	0.850	0.137	0.126	0.889
	**2011**	**2012**	**2013**	**2014**	**2015**
**AU**	14.74	19.31	20.46	19.06	18.13
**AM**	14.69	17.39	19.25	18.46	17.89
**EU**	14.41	18.14	21.25	19.73	18.14
**EM**	14.33	18.10	17.73	18.31	17.38
**M**	0.927	0.453	0.004	0.247	0.725
**CO_2_**	0.529	0.765	0.649	0.760	0.761
**MxCO_2_**	0.984	0.214	0.149	0.635	0.755

^a^ Treatments = ambient CO_2_-unmanaged (AU); ambient CO_2_-managed (AM); elevated CO_2_-unmanaged (EU); elevated CO_2_-managed (EM). ^b^ *p* values for the main effects of management (M), carbon dioxide (CO_2_), and their interaction (MxCO_2_).

**Table 4 plants-13-00485-t004:** Bahiagrass root length density (km m^−3^) by depth for each of the 10 years of study. Data shown are means (N = 3) with statistics.

	**2006**					
**Trt** ^**a**^	**0–5 cm**	**5–10 cm**	**10–15 cm**	**15–30 cm**	**30–45 cm**	**45–60 cm**
**AU**	123.89	60.25	42.03	34.58	37.62	26.89
**AM**	217.79	126.09	73.48	50.50	48.74	35.76
**EU**	132.11	59.08	38.06	33.90	35.12	26.06
**EM**	228.22	125.21	91.70	55.45	45.66	40.56
**M ^b^**	0.007	0.021	0.026	0.019	0.015	0.015
**CO_2_**	0.732	0.837	0.383	0.501	0.448	0.615
**MxCO_2_**	0.968	0.976	0.202	0.385	0.936	0.480
	**2007**					
**Trt ^a^**	**0–5 cm**	**5–10 cm**	**10–15 cm**	**15–30 cm**	**30–45 cm**	**45–60 cm**
**AU**	182.96	42.91	33.36	28.71	28.22	34.39
**AM**	339.18	144.02	84.65	75.44	52.17	55.65
**EU**	217.65	53.20	41.15	34.44	32.28	30.72
**EM**	322.57	112.28	93.02	63.44	43.65	46.29
**M**	0.002	0.001	0.002	0.009	0.019	0.034
**CO_2_**	0.737	0.246	0.098	0.328	0.343	0.030
**MxCO_2_**	0.356	0.056	0.942	0.035	0.039	0.226
	**2008**					
**Trt ^a^**	**0–5 cm**	**5–10 cm**	**10–15 cm**	**15–30 cm**	**30–45 cm**	**45–60 cm**
**AU**	256.44	113.45	61.87	62.06	55.50	49.62
**AM**	287.59	214.41	108.60	97.68	69.66	80.88
**EU**	300.73	134.17	70.10	64.12	56.23	54.18
**EM**	257.91	239.98	118.45	98.12	64.71	71.72
**M**	0.824	<0.001	0.032	0.059	0.051	0.007
**CO_2_**	0.779	0.032	0.262	0.885	0.681	0.616
**MxCO_2_**	0.178	0.781	0.913	0.925	0.581	0.182
	**2009**					
**Trt ^a^**	**0–5 cm**	**5–10 cm**	**10–15 cm**	**15–30 cm**	**30–45 cm**	**45–60 cm**
**AU**	339.91	81.27	49.23	50.06	46.63	38.85
**AM**	445.13	181.34	113.16	95.23	69.21	89.16
**EU**	365.04	82.88	52.76	45.21	43.55	37.82
**EM**	491.13	196.04	149.02	120.80	75.93	66.82
**M**	0.012	<0.001	<0.001	0.002	0.052	0.003
**CO_2_**	0.318	0.552	0.054	0.277	0.857	0.013
**MxCO_2_**	0.760	0.632	0.100	0.138	0.632	0.018
	**2010**					
**Trt ^a^**	**0–5 cm**	**5–10 cm**	**10–15 cm**	**15–30 cm**	**30–45 cm**	**45–60 cm**
**AU**	315.81	94.20	78.77	49.87	47.61	49.87
**AM**	409.13	186.93	137.99	102.28	79.60	112.86
**EU**	318.75	97.73	69.36	46.68	48.25	43.94
**EM**	420.45	184.43	136.97	96.50	79.01	93.27
**M**	0.051	<0.001	0.001	<0.001	0.003	0.029
**CO_2_**	0.798	0.944	0.615	0.457	0.998	0.431
**MxCO_2_**	0.880	0.686	0.685	0.827	0.938	0.664
	**2011**					
**Trt ^a^**	**0–5 cm**	**5–10 cm**	**10–15 cm**	**15–30 cm**	**30–45 cm**	**45–60 cm**
**AU**	406.48	95.23	64.07	48.40	49.57	40.41
**AM**	441.90	231.01	130.35	99.39	85.19	73.48
**EU**	415.45	103.31	71.57	43.06	41.10	39.04
**EM**	433.38	242.33	165.77	122.07	102.63	104.00
**M**	0.661	<0.001	<0.001	0.005	0.005	0.003
**CO_2_**	0.995	0.363	0.024	0.278	0.340	0.035
**MxCO_2_**	0.808	0.873	0.107	0.112	0.035	0.027
	**2012**					
**Trt ^a^**	**0–5 cm**	**5–10 cm**	**10–15 cm**	**15–30 cm**	**30–45 cm**	**45–60 cm**
**AU**	436.76	117.27	52.61	42.52	39.63	40.80
**AM**	407.95	241.01	140.49	101.11	77.30	92.29
**EU**	495.54	104.49	69.95	49.23	43.55	41.05
**EM**	416.48	204.42	110.07	96.65	69.12	92.98
**M**	0.381	<0.001	0.013	0.026	0.010	0.008
**CO_2_**	0.317	0.120	0.423	0.776	0.827	0.975
**MxCO_2_**	0.442	0.427	0.031	0.206	0.538	0.988
	**2013**					
**Trt ^a^**	**0–5 cm**	**5–10 cm**	**10–15 cm**	**15–30 cm**	**30–45 cm**	**45–60 cm**
**AU**	341.82	87.73	63.63	44.13	41.64	37.28
**AM**	390.17	214.56	115.36	85.19	57.26	97.53
**EU**	350.05	68.77	52.17	36.94	34.59	31.35
**EM**	378.42	199.57	109.78	98.12	74.17	105.07
**M**	0.260	<0.001	0.002	0.002	0.001	0.001
**CO_2_**	0.956	0.437	0.494	0.789	0.409	0.954
**MxCO_2_**	0.757	0.926	0.811	0.364	0.067	0.635
	**2014**					
**Trt ^a^**	**0–5 cm**	**5–10 cm**	**10–15 cm**	**15–30 cm**	**30–45 cm**	**45–60 cm**
**AU**	400.16	118.89	50.11	35.57	29.05	30.08
**AM**	386.94	191.19	90.52	69.56	52.07	86.02
**EU**	379.15	82.73	47.17	31.79	32.72	38.95
**EM**	390.61	175.61	78.62	73.82	58.44	87.98
**M**	0.981	0.005	0.003	0.024	0.039	0.080
**CO_2_**	0.716	0.221	0.408	0.938	0.281	0.328
**MxCO_2_**	0.607	0.607	0.612	0.243	0.755	0.517
	**2015**					
**Trt ^a^**	**0–5 cm**	**5–10 cm**	**10–15 cm**	**15–30 cm**	**30–45 cm**	**45–60 cm**
**AU**	341.53	89.20	53.49	36.84	38.36	41.00
**AM**	333.45	191.78	89.64	79.80	76.96	42.81
**EU**	399.43	94.34	64.22	39.39	39.97	101.89
**EM**	432.94	192.15	78.03	64.91	63.93	67.75
**M**	0.569	<0.001	0.170	0.105	0.189	0.015
**CO_2_**	0.010	0.829	0.958	0.464	0.544	0.254
**MxCO_2_**	0.362	0.871	0.225	0.317	0.443	0.210

^a^ Treatments = ambient CO_2_-unmanaged (AU); ambient CO_2_-managed (AM); elevated CO_2_-unmanaged (EU); elevated CO_2_-managed (EM). ^b^ *p* values for the main effects of management (M), carbon dioxide (CO_2_), and their interaction (MxCO_2_).

**Table 5 plants-13-00485-t005:** Bahiagrass root weight density (kg m^−3^) by depth for each of the 10 years of study. Data shown are means (N = 3) with statistics.

	**2006**					
**Trt** ^**a**^	**0–5 cm**	**5–10 cm**	**10–15 cm**	**15–30 cm**	**30–45 cm**	**45–60 cm**
**AU**	1.36	0.92	0.64	0.48	0.51	0.47
**AM**	3.15	2.19	1.07	0.70	0.66	0.63
**EU**	1.54	0.89	0.64	0.49	0.57	0.43
**EM**	3.09	2.05	1.37	0.80	0.65	0.75
**M ^b^**	0.001	0.011	0.006	0.007	0.043	0.019
**CO_2_**	0.871	0.494	0.372	0.357	0.660	0.639
**MxCO_2_**	0.745	0.670	0.359	0.389	0.478	0.359
	**2007**					
**Trt ^a^**	**0–5 cm**	**5–10 cm**	**10–15 cm**	**15–30 cm**	**30–45 cm**	**45–60 cm**
**AU**	2.03	0.66	0.47	0.39	0.43	0.57
**AM**	4.23	2.23	1.50	1.21	0.97	1.14
**EU**	2.00	0.73	0.60	0.49	0.43	0.55
**EM**	3.87	1.98	1.43	1.07	0.67	0.98
**M**	0.001	0.019	<0.001	0.001	0.076	0.013
**CO_2_**	0.579	0.447	0.828	0.740	0.122	0.226
**MxCO_2_**	0.637	0.202	0.469	0.156	0.110	0.337
	**2008**					
**Trt ^a^**	**0–5 cm**	**5–10 cm**	**10–15 cm**	**15–30 cm**	**30–45 cm**	**45–60 cm**
**AU**	3.10	1.38	0.93	0.85	0.86	0.93
**AM**	3.59	2.75	1.58	1.43	1.24	1.56
**EU**	3.24	1.33	0.96	0.86	0.86	0.91
**EM**	3.31	3.26	1.92	1.58	1.14	1.33
**M**	0.495	0.001	0.001	0.006	0.005	0.038
**CO_2_**	0.854	0.217	0.292	0.454	0.568	0.126
**MxCO_2_**	0.602	0.154	0.357	0.521	0.536	0.185
	**2009**					
**Trt ^a^**	**0–5 cm**	**5–10 cm**	**10–15 cm**	**15–30 cm**	**30–45 cm**	**45–60 cm**
**AU**	2.49	0.90	0.55	0.61	0.60	0.53
**AM**	4.51	2.26	1.48	1.35	1.15	1.51
**EU**	2.89	0.95	0.68	0.59	0.59	0.54
**EM**	4.58	2.40	2.10	1.78	1.03	1.02
**M**	0.006	0.001	<0.001	0.010	0.010	0.002
**CO_2_**	0.483	0.224	0.009	0.016	0.667	0.015
**MxCO_2_**	0.615	0.562	0.036	0.011	0.717	0.012
	**2010**					
**Trt ^a^**	**0–5 cm**	**5–10 cm**	**10–15 cm**	**15–30 cm**	**30–45 cm**	**45–60 cm**
**AU**	2.82	0.95	0.81	0.63	0.69	0.71
**AM**	4.84	2.68	1.97	1.54	1.39	1.94
**EU**	2.99	1.04	0.85	0.60	0.59	0.61
**EM**	4.77	2.50	1.98	1.60	1.29	1.49
**M**	0.006	<0.001	<0.001	<0.001	0.001	0.001
**CO_2_**	0.852	0.739	0.831	0.904	0.421	0.209
**MxCO_2_**	0.692	0.362	0.880	0.702	0.992	0.408
	**2011**					
**Trt ^a^**	**0–5 cm**	**5–10 cm**	**10–15 cm**	**15–30 cm**	**30–45 cm**	**45–60 cm**
**AU**	4.17	1.39	0.86	0.82	0.83	0.74
**AM**	5.76	3.34	1.65	1.38	1.43	1.25
**EU**	4.80	1.39	0.86	0.62	0.72	0.64
**EM**	6.34	3.48	2.20	1.58	1.47	1.69
**M**	0.054	0.020	<0.001	0.014	0.007	0.025
**CO_2_**	0.050	0.811	0.011	0.882	0.451	0.166
**MxCO_2_**	0.903	0.795	0.012	0.205	0.142	0.059
	**2012**					
**Trt ^a^**	**0–5 cm**	**5–10 cm**	**10–15 cm**	**15–30 cm**	**30–45 cm**	**45–60 cm**
**AU**	6.28	2.31	1.04	0.88	0.92	0.85
**AM**	6.35	3.71	1.85	1.82	1.73	1.90
**EU**	7.96	2.08	1.10	0.89	0.88	0.79
**EM**	6.47	3.49	1.68	1.59	1.33	1.96
**M**	0.270	0.030	0.010	0.085	0.030	0.014
**CO_2_**	0.099	0.528	0.795	0.631	0.217	0.994
**MxCO_2_**	0.138	0.991	0.579	0.597	0.293	0.869
	**2013**					
**Trt ^a^**	**0–5 cm**	**5–10 cm**	**10–15 cm**	**15–30 cm**	**30–45 cm**	**45–60 cm**
**AU**	5.29	1.67	1.19	0.88	0.86	0.99
**AM**	6.15	3.48	2.00	1.87	1.37	2.10
**EU**	5.76	1.52	1.12	0.76	0.86	0.71
**EM**	5.98	3.78	1.85	1.72	1.44	1.85
**M**	0.219	0.002	0.037	0.001	0.062	0.011
**CO_2_**	0.721	0.853	0.467	0.416	0.816	0.459
**MxCO_2_**	0.449	0.600	0.790	0.946	0.812	0.971
	**2014**					
**Trt ^a^**	**0–5 cm**	**5–10 cm**	**10–15 cm**	**15–30 cm**	**30–45 cm**	**45–60 cm**
**AU**	5.13	1.76	1.10	0.73	0.68	0.67
**AM**	6.26	3.32	1.91	1.31	1.02	1.46
**EU**	5.07	1.31	1.35	0.73	0.70	0.69
**EM**	6.21	2.97	1.58	1.38	1.18	1.64
**M**	0.058	0.105	0.015	<0.001	0.094	0.028
**CO_2_**	0.710	0.216	0.836	0.661	0.556	0.371
**MxCO_2_**	0.980	0.876	0.126	0.651	0.615	0.459
	**2015**					
**Trt ^a^**	**0–5 cm**	**5–10 cm**	**10–15 cm**	**15–30 cm**	**30–45 cm**	**45–60 cm**
**AU**	4.31	1.50	0.96	0.74	0.72	0.78
**AM**	4.99	2.89	1.78	1.48	1.54	1.91
**EU**	4.75	1.69	1.03	0.79	0.82	0.75
**EM**	5.46	2.88	1.38	1.24	1.44	1.19
**M**	0.108	<0.001	0.017	0.040	0.026	0.013
**CO_2_**	0.267	0.685	0.330	0.468	0.983	0.146
**MxCO_2_**	0.975	0.658	0.187	0.297	0.719	0.171

^a^ Treatments = ambient CO_2_-unmanaged (AU); ambient CO_2_-managed (AM); elevated CO_2_-unmanaged (EU); elevated CO_2_-managed (EM). ^b^ *p* values for the main effects of management (M), carbon dioxide (CO_2_), and their interaction (MxCO_2_).

**Table 6 plants-13-00485-t006:** Bahiagrass lineal root density (g km^−1^) by depth for each of the 10 years of study. Data shown are means (N = 3) with statistics.

	**2006**					
**Trt** ^**a**^	**0–5 cm**	**5–10 cm**	**10–15 cm**	**15–30 cm**	**30–45 cm**	**45–60 cm**
**AU**	11.01	15.72	16.15	14.00	13.60	17.48
**AM**	15.04	17.57	15.03	14.32	13.60	17.88
**EU**	11.85	15.06	17.31	14.44	16.37	16.31
**EM**	13.36	16.37	14.84	14.44	14.30	18.56
**M ^b^**	0.115	0.393	0.450	0.897	0.326	0.462
**CO_2_**	0.644	0.237	0.834	0.806	0.122	0.857
**MxCO_2_**	0.211	0.707	0.772	0.892	0.325	0.510
	**2007**					
**Trt ^a^**	**0–5 cm**	**5–10 cm**	**10–15 cm**	**15–30 cm**	**30–45 cm**	**45–60 cm**
**AU**	11.40	16.18	14.70	13.92	15.34	16.61
**AM**	12.70	15.66	17.95	16.37	18.34	20.42
**EU**	9.08	14.00	14.72	14.49	13.54	17.98
**EM**	11.86	17.68	15.49	16.77	15.27	21.26
**M**	0.057	0.567	0.428	0.224	0.132	0.034
**CO_2_**	0.123	0.955	0.627	0.779	0.117	0.360
**MxCO_2_**	0.443	0.177	0.619	0.960	0.631	0.816
	**2008**					
**Trt ^a^**	**0–5 cm**	**5–10 cm**	**10–15 cm**	**15–30 cm**	**30–45 cm**	**45–60 cm**
**AU**	12.95	12.32	15.48	13.88	15.50	18.45
**AM**	13.17	13.44	14.59	14.80	17.87	19.36
**EU**	11.01	10.05	14.36	13.64	15.68	16.98
**EM**	15.64	13.71	16.19	16.52	17.58	18.56
**M**	0.218	0.178	0.750	0.148	0.205	0.622
**CO_2_**	0.885	0.316	0.872	0.539	0.971	0.516
**MxCO_2_**	0.256	0.218	0.372	0.424	0.871	0.845
	**2009**					
**Trt ^a^**	**0–5 cm**	**5–10 cm**	**10–15 cm**	**15–30 cm**	**30–45 cm**	**45–60 cm**
**AU**	7.30	11.40	11.47	12.23	12.69	16.69
**AM**	10.31	12.63	13.22	14.25	16.94	16.88
**EU**	8.00	11.60	12.89	13.57	13.64	14.26
**EM**	9.36	12.26	14.14	14.92	13.23	15.53
**M**	0.126	0.259	0.256	0.336	0.178	0.056
**CO_2_**	0.842	0.918	0.180	0.408	0.320	0.704
**MxCO_2_**	0.225	0.716	0.742	0.771	0.111	0.365
	**2010**					
**Trt ^a^**	**0–5 cm**	**5–10 cm**	**10–15 cm**	**15–30 cm**	**30–45 cm**	**45–60 cm**
**AU**	8.99	10.16	10.52	12.68	14.24	14.94
**AM**	12.02	14.36	14.52	15.00	17.87	17.18
**EU**	9.36	10.53	12.38	12.76	12.14	13.73
**EM**	11.42	13.56	14.72	16.64	16.31	16.82
**M**	0.018	0.004	0.019	0.138	0.032	0.209
**CO_2_**	0.900	0.732	0.340	0.260	0.240	0.700
**MxCO_2_**	0.586	0.374	0.436	0.300	0.856	0.833
	**2011**					
**Trt ^a^**	**0–5 cm**	**5–10 cm**	**10–15 cm**	**15–30 cm**	**30–45 cm**	**45–60 cm**
**AU**	10.14	14.39	13.44	16.06	16.85	18.06
**AM**	13.28	14.57	12.66	13.84	16.80	17.08
**EU**	11.68	13.47	12.57	15.10	17.28	17.05
**EM**	14.56	14.28	13.36	12.96	14.58	16.20
**M**	0.005	0.688	0.999	0.204	0.330	0.709
**CO_2_**	0.089	0.473	0.940	0.048	0.178	0.701
**MxCO_2_**	0.860	0.699	0.500	0.477	0.073	0.978
	**2012**					
**Trt ^a^**	**0–5 cm**	**5–10 cm**	**10–15 cm**	**15–30 cm**	**30–45 cm**	**45–60 cm**
**AU**	14.63	19.00	19.54	20.17	23.21	20.72
**AM**	15.68	15.27	13.13	17.34	22.28	20.43
**EU**	16.09	20.01	15.74	17.91	20.31	19.38
**EM**	16.08	17.23	16.25	16.92	20.79	21.44
**M**	0.786	0.084	0.131	0.428	0.928	0.603
**CO_2_**	0.625	0.393	0.848	0.561	0.377	0.922
**MxCO_2_**	0.780	0.779	0.086	0.688	0.766	0.494
	**2013**					
**Trt ^a^**	**0–5 cm**	**5–10 cm**	**10–15 cm**	**15–30 cm**	**30–45 cm**	**45–60 cm**
**AU**	15.71	18.80	20.48	21.00	20.89	27.09
**AM**	15.81	16.39	17.43	21.84	23.23	20.87
**EU**	16.60	22.22	22.02	20.76	25.07	22.38
**EM**	15.82	18.81	16.7	17.66	19.56	18.05
**M**	0.850	0.033	0.158	0.514	0.533	0.060
**CO_2_**	0.512	0.032	0.882	0.219	0.911	0.156
**MxCO_2_**	0.524	0.671	0.679	0.269	0.142	0.705
	**2014**					
**Trt ^a^**	**0–5 cm**	**5–10 cm**	**10–15 cm**	**15–30 cm**	**30–45 cm**	**45–60 cm**
**AU**	12.92	15.35	21.96	20.44	23.50	22.46
**AM**	16.43	17.22	21.14	19.38	19.77	17.37
**EU**	13.40	16.51	28.97	22.73	21.49	17.80
**EM**	15.94	16.49	20.14	18.76	19.64	19.01
**M**	0.012	0.628	0.004	0.250	0.179	0.442
**CO_2_**	0.999	0.909	0.021	0.526	0.588	0.480
**MxCO_2_**	0.619	0.620	0.009	0.292	0.631	0.180
	**2015**					
**Trt ^a^**	**0–5 cm**	**5–10 cm**	**10–15 cm**	**15–30 cm**	**30–45 cm**	**45–60 cm**
**AU**	12.59	16.82	18.81	20.87	20.70	20.79
**AM**	14.92	15.09	20.03	18.66	20.31	18.69
**EU**	12.02	18.01	17.24	21.32	22.49	20.01
**EM**	12.72	14.92	17.89	19.24	22.94	17.54
**M**	0.206	0.037	0.762	0.375	0.994	0.437
**CO_2_**	0.245	0.254	0.364	0.336	0.271	0.736
**MxCO_2_**	0.480	0.152	0.884	0.896	0.821	0.949

^a^ Treatments = ambient CO_2_-unmanaged (AU); ambient CO_2_-managed (AM); elevated CO_2_-unmanaged (EU); elevated CO_2_-managed (EM). ^b^
*p* values for the main effects of management (M), carbon dioxide (CO_2_), and their interaction (MxCO_2_).

**Table 7 plants-13-00485-t007:** Bahiagrass rhizome biomass and root-to-shoot ratio at study termination. Data shown are means (N = 3) with statistics.

Trt ^a^	Rhizomes (kg ha^−1^)	Root:Shoot ^c^
**AU**	4496.2	43.52
**AM**	8167.2	18.29
**EU**	5582.2	45.06
**EM**	8751.4	14.15
**M ^b^**	<0.001	0.004
**CO_2_**	0.136	0.725
**MxCO_2_**	0.602	0.457

^a^ Treatments = ambient CO_2_-unmanaged (AU); ambient CO_2_-managed (AM); elevated CO_2_-unmanaged (EU); elevated CO_2_-managed (EM). ^b^p values for the main effects of management (M), carbon dioxide (CO_2_), and their interaction (MxCO_2_). ^c^ Root-to-shoot ratio = root + rhizome dry weights at study termination divided by the 2015 final aboveground dry weight previously reported by Prior et al. [23].

## Data Availability

Data will be made available upon request.

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
