# Peer review of "Belowground Response of a Bahiagrass Pasture to Long-Term Elevated [CO2] and Soil Fertility Management"

_plants, 2024, doi:10.3390/plants13040485_

Round 1

Reviewer 1 Report

Comments and Suggestions for Authors

Dear Authors,

In view of mine manuscript falls under the scope of the journal and has potential interest to the global researcher. At present, structure of the article is the main problem, reset things looks good to me. 

State of the art needs to be improved, kindly see the suggested article to improve it  (remove the citations 35-37, its too old, and replace them with the recent ones to make the statement robust).....https://doi.org/10.1016/j.catena.2021.105667 ;  https://doi.org/10.1002/ldr.4307 ; https://doi.org/10.3389/fenvs.2021.724950

atmospheric CO2 concentration [CO2] or atmospheric CO2 ?? i don't understand the logic behind writing [CO2] 

Kindly mention the name of a few C3/C4 plants in brackets to make the reader clear about it  ...

Authors jumped directly from the introduction to the result, and discussion, authors need to follow the structure (Introduction> research gaps> objective> material and method section > result and discussion (separately of combined)> conclusion and future recommendation

Authors prepared many tables, so I suggested making some figures instead of many tables (3-6) to make the representation good, please 

Conclusion is very very short and generalized, and needs to be improved a lot as this part is very important to the reader 

Comments on the Quality of English Language

English is written in scientific ways, so expert editing is required 

Author Response

Please see attachment. Author comments follow each reviewer comment and are highlighted in yellow.

Reviewer 2 Report

Comments and Suggestions for Authors

General comments

Good paper. I congratulate the authors for publishing the root data from this long-term study. 

A couple of graphs would be helpful to the reader.

Below are a few minor comments on the manuscript  

Line 16  Lime is not a fertilizer, so change to fertilizer and lime treatment

For in all the tables clarify if the data are means or averages.

Line 182 The authors indicate it is unexpected and then make the case for why it would be expected.  

Line 230-242.  Need to clarify for the reader the discussion on the root to shoot ratio. One could question the use of the above ground biomass for October only, when the below ground biomass would more likely be the product of growth throughout the previous 12 months, given it includes the rhizomes. On that basis the S:R ratio should show a decline, given the above grow response to CO2.

There are few edits on the manuscript the authors might like to consider. 

Author Response

Please see attachment. Author responses to reviewer comments follow each comment and are highlighted in yellow.

Round 2

Reviewer 1 Report

Comments and Suggestions for Authors

Dear authors, 

In view of mine, you have not carefully read the article and scientific way of writing, kindly see some published papers and correct the format as suggested:- Introduction > Methodologies (study area, methodologies applied, and statistics used ) > Results> Discussion (including implementation or shortcomings of the research work) > Conclusion 

Abbreviations need to be defined (line no 95) first before being used for the first time in the abstract 

Some of the tables could be reduced and making them into figures would be good for scientific representations, kindly revise 

Comments on the Quality of English Language

ok ok..need minor corrections only but can be done in proofreading